# An evaluation of fusion partner proteins for paratransgenesis in *Asaia bogorensis*

**Christina Grogan**[ID]**, Marissa Bennett, David J. Lampe**[ID] *

Department of Biological Sciences, Bayer School of Natural and Environmental Sciences, Duquesne University, Pittsburgh, PA, United States of America

* lampe@duq.edu

**Data Availability Statement:** All figure files are available from the Dryad database (doi:10.5061/dryad.n2z34tmxw). Plasmid sequences are available in genbank (accession number(s) MZ215986 to MZ215990).

## Abstract

Mosquitoes transmit many pathogens responsible for human diseases, such as malaria which is caused by parasites in the genus *Plasmodium*. Current strategies to control vector-transmitted diseases are increasingly undermined by mosquito and pathogen resistance, so additional methods of control are required. Paratransgenesis is a method whereby symbiotic bacteria are genetically modified to affect the mosquito's phenotype by engineering them to deliver effector molecules into the midgut to kill parasites. One paratransgenesis candidate is *Asaia bogorensis*, a Gram-negative bacterium colonizing the midgut, ovaries, and salivary glands of *Anopheles* sp. mosquitoes. Previously, engineered *Asaia* strains using native signals to drive the release of the antimicrobial peptide, scorpine, fused to alkaline phosphatase were successful in significantly suppressing the number of oocysts formed after a blood meal containing *P. berghei*. However, these strains saw high fitness costs associated with the production of the recombinant protein. Here, we report evaluation of five different partner proteins fused to scorpine that were evaluated for effects on the growth and fitness of the transgenic bacteria. Three of the new partner proteins resulted in significant levels of protein released from the *Asaia* bacterium while also significantly reducing the prevalence of mosquitoes infected with *P. berghei*. Two partners performed as well as the previously tested *Asaia* strain that used alkaline phosphatase in the fitness analyses, but neither exceeded it. It may be that there is a maximum level of fitness and parasite inhibition that can be achieved with scorpine being driven constitutively, and that use of a *Plasmodium* specific effector molecule in place of scorpine would help to mitigate the stress on the symbionts.

## Introduction

Mosquitoes threaten global public health by vectoring many human pathogens. These vector-borne diseases include dengue, Zika virus fever, yellow fever, and malaria, representing more than 17% of all infectious diseases and over 700,000 deaths per year [1]. Malaria, caused by *Plasmodium* parasites transmitted by *Anopheles* mosquitoes, accounts for over 219 million infections and over 400,000 deaths annually [2]. Current preventative strategies focus on control of the parasites at the blood stage through antimalarial drugs, or use insecticides to reduce

**Funding:** This work was supported by National Institutes of Health award 2R15 AI107735 (www. nih.gov) to DL. The funders had no role in study design, data collection and analysis, decision to publish, or preparation of the manuscript.

**Competing interests:** The authors have declared that no competing interests exist.

mosquito populations to prevent transmission of the disease. In recent years, concerns have been raised about potential hazards from insecticides on human and environmental health [3]. In addition, insects and parasites have both evolved resistance to their respective control measures, reducing the effectiveness of these existing strategies [4, 5]. Due to these challenges, alternative and sustainable new tactics to combat this disease are necessary.

Genetic control methods for mosquitoes recently have gained much attention. One such method focuses on mosquito population suppression by introducing genes to cause sterility and death in natural populations [6–8]. Another strategy aims to reduce the insect vectorial capacity through population replacement with mosquitoes engineered with genetic traits to interfere with the host life cycle of the pathogen [9].While RIDL (Release of Insects carrying a Dominant Lethal) has recently proven successful at population suppression in *Aedes* sp. and some *Anopheles* sp. mosquitoes, field applications of these strategies might still prove challenging due to the large number of *Anopheles* species that can vector the pathogens, reproductively isolated populations, and the maintenance required for the transgenes to spread and be sustained in large populations [10–12]. Another proposed genetic method focuses on the symbiotic microorganisms that reside within the insect hosts. *Paratransgenesis* is the use of symbionts to express and deliver antimalarial effector molecules to kill the parasite in the mosquito host. Mosquitoes are hosts to a variety of microbial communities, consisting of bacteria, fungi, and viruses, all of which are currently being explored for their ability for vector control and pathogen transmission control [13–15]. One particular bacterium, *Asaia bogorensis* SF2.1, colonizes a number of mosquito (= Culicidae) species that feed on the nectar of plants, including *Anopheles stephensi*, *An. gambiae*, *An. maculipennis*, *Aedes aegypti*, *Ae. albopictus*, and *Culex pipiens*, and other arthropods such as *Scaphoideus titanus* (Cicadellidae), and *Sogatella furcifera* (Delphacidae), most of which can vector human or plant diseases [16–21]. *Asaia* spreads horizontally and vertically through mosquito populations, colonizing the ovaries, testes, salivary glands, and the midgut tissues [16, 22, 23]. *Asaia* is amenable to genetic engineering. Importantly, bacteria in this genus are reported to cause very few human infections [24–30].

We recently demonstrated the successful release of scorpine from *Asaia* SF2.1 into the midgut for *Plasmodium* inhibition using native *Asaia SF2.1* signal sequences with alkaline phosphatase fused to the C-terminus of scorpine [31]. Isolated from the venom of the scorpion *Pandinus imperator*, the antimalarial effector molecule scorpine is a 75 amino acid peptide that has antimicrobial activity [32, 33]. This effector is also a potent inhibitor of *Plasmodium* oocyst development in anopheline mosquito midguts [34–36]. Expression of scorpine in *Asaia*, however, leads to strains with significantly lower fitness than wild type strains, so there is room to improve the expression of this antiplasmodial via other arrangements [31].

Decreased bacterial fitness has been shown to result from the overexpression of proteins in other bacterial species such as *E. coli* [37–39]. The native form of the effector molecule bee venom PLA2 could not be expressed in a similar paratransgenic study because it caused mortality of the transgenic bacteria [40]. Production of these effectors not only stress the organism, but since some of these peptides are antibacterial in nature, they can be toxic to the host cell. Many researchers have worked on the problem by fusing antimicrobial peptides to other partner or carrier proteins. One such study used the catalytic domain of a cellulase KSM-64 from *Bacillus sp*. as a fusion protein, which was shown to allow high amounts of antimicrobial peptides to be secreted without any fitness costs to the *E. coli* host [41]. Fusion expression to partner or carrier proteins is important for multiple reasons: the partner can prevent toxicity of the antimalarials in the bacterial host cell, they can stabilize the construct and/or assist in folding of the peptide, and they may protect the recombinant protein from intracellular proteolysis. Most partner proteins are heterologous and have become useful in recombinant peptide

applications. Common partner proteins include maltose binding protein (MBP), thioredoxin, and glutathione-S- transferase (GST), which promote the soluble expression of recombinant peptides in *E. coli* [42–44]. Simple fusion partners such as a 6-His with a linker region have also been used [45–47].

While these studies were conducted for clinical and industrial purposes, the application of partner proteins fused to the antimalarial peptides within *Asaia* seems attractive as well. In previous studies, alkaline phosphatase proved to be a useful fusion partner; however, this protein is relatively large and subject to cleavage [31]. We reasoned that using different fusion partners that could assist in the folding and the solubility of the recombinant protein would also have a lesser impact on the fitness of the *Asaia* host. We report here the evaluation of five different tags or partner proteins fused to the strong antimalarial effector molecule scorpine.

## Materials and methods

### Media and antibiotics

For plasmid cloning, *E. coli* Top10F' cells were cultured using standard Luria Bertani (LB) broth (1% Tryptone, 0.5% NaCl, 0.5% yeast extract (w/v)) and LB agar (LB broth with 15 g/L agar). Media was supplemented with 30 μg/mL kanamycin. All *Asaia* strains were cultured in mannitol broth (0.5% yeast extract, 0.3% peptone, 2.5% mannitol (w/v)) or mannitol agar (mannitol broth with 15 g/L agar), both adjusted to a pH of 6.5 before sterilization. Davis minimal media broth (0.7% dipotassium phosphate, 0.2% monopotassium phosphate, 0.05% sodium citrate, 0.01% magnesium sulfate, 0.1% ammonium sulfate (w/v)) was used for cell collection and Davis minimal media agar (minimal broth with 15 g/L agar) supplemented with 0.5% (w/v) arabinose solution was used for colonization assessments. Media was supplemented with 120 μg/mL kanamycin for plasmid selection and also 100 μg/mL ampicillin for colonization assessments. Both liquid and solid media cultures for all strains were grown at 30˚C, with agitation for liquid cultures. All bacterial strains and plasmids used in this study are described in Table 1.

### Mosquito and parasite maintenance

*Anopheles stephensi* (a gift from the Johns Hopkins Malaria Research Institute) were maintained on 10% (w/v) sucrose solution at 29˚C and 70% humidity with a 12h day:12h night light cycle. Larvae were reared at 29˚C in pans of water and fed on crushed Tetramin Tropical Tablets for Bottom Feeders. Pupae were collected by hand and allowed to emerge as adults in 0.03 m$^3$ screened cages. *Plasmodium berghei* strain ANKA2.34 was maintained by passage through 7–8 week-old outbred female ND4 Swiss Webster mice (Charles River Laboratory) using standard procedures [52]. This study was carried out in strict accordance with the recommendations in the Guide for the Care and Use of Laboratory Animals of the National Institutes of Health and Duquesne University IACUC protocol #1810–09. All surgery was performed using anesthesia as outlined below, and all efforts were made to minimize suffering.

### Plasmid construction

All plasmid construction and propagation was performed in *E. coli* Top10F' (Invitrogen). Plasmid sequences for all plasmids constructed for this study were deposited into Genbank under accession numbers MZ215986 to MZ215990. The pHyp4s-PhoA (hereafter PhoA) vector was used for construction of pHyp4s.Myc (hereafter Myc), which was then used for all *Asaia sp*. fusion partner plasmids (Fig 1) [31]. This plasmid uses the pBBR broad-host range origin of replication, a constitutively active neomycin phosphotransferase promoter (P$_{nptII}$), and the

**Table 1. Strains and plasmids used in this study.**

| Strains | Characteristics | Reference |
|---|---|---|
| *E. coli* Top10F' | F´{*lacIq*, Tn*10*(TetR)} *mcrA* Δ(*mrr-hsdRMS-mcrBC*) Φ80*lacZΔM15 ΔlacX74 recA1 araD139 Δ(ara leu) 7697 galU galK rpsL* (StrR) *endA1 nupG* | [48] |
| *Asaia* SF2.1 | Wild type strain isolated from *Anopheles* mosquitoes | [16] |
| *E. coli* pIT2-scFv | Amp$^R$, colE1 ori, P$_{lac}$ promoter, pelB leader, anti-BSA, myc tag | [49] |
| **Plasmids** | **Relevant Characteristics** | **Reference** |
| pMALc2X | Source for *malE* gene that encodes for maltose binding protein (MBP) | New England Biolabs, USA |
| pET-32a | Source for *trxA* gene that encodes thioredoxin protein (TrxA) | [50] |
| pGEX-4T-1 | Source for *gst-3* gene that encodes glutathione s-transferase protein (GST) | [51] |
| pHyp4 | Kan$^R$, pBBR ori, P$_{nptII}$ promoter, Hypothetical Protein 4 signal- PhoA construct. | [31] |
| pHyp4s | Kan$^R$, pBBR ori, P$_{nptII}$ promoter, Hypothetical Protein 4 signal-scorpine-(GGGGS)$_3$- PhoA effector construct. Also used for plasmid construction. | [31] |
| pHyp4s.Myc[1] | Hypothetical Protein 4 signal-scorpine-(GGGGS)$_3$- Myc effector construct. Also used for plasmid construction. | This study |
| pHyp4s.TrxA | Hypothetical Protein 4 signal-scorpine-(GGGGS)$_3$- TrxA effector construct | This study |
| pHyp4s.GST | Hypothetical Protein 4 signal-scorpine-(GGGGS)$_3$- GST effector construct | This study |
| pHyp4s.MBP | Hypothetical Protein 4 signal-scorpine-(GGGGS)$_3$- MBP effector construct | This study |
| pHyp4s.His | Hypothetical Protein 4 signal-scorpine-(GGGGS)$_3$- 6xHis effector construct | This study |

Transgenic *Asaia* strains are referred to as the fusion partner protein names.

[1] Plasmid sequences for plasmids constructed for this study were deposited into Genbank under accession numbers MZ215986 to MZ215990.

neomycin phosphotransferase II gene (*nptII*) conferring kanamycin resistance. An epitope tag followed by a multiple cloning site was inserted after the (Gly-Gly-Gly-Gly-Ser)$_3$ flexible linker using standard Gibson assembly procedures [53], replacing the alkaline phosphatase reporter gene. Briefly, the PhoA plasmid was amplified using the secreted scorpine vector forward and reverse primers (S1 Table). After size confirmation using gel electrophoresis, a PCR cleanup was performed (Zymo, #D4029). A Gblock (IDT) synthetic dsDNA fragment with homologous regions to the amplified PhoA vector was designed with a Myc epitope tag followed by *AvrII* and *AscI* restriction digestion sites and an in frame stop codon (S1 Table). A two-part Gibson assembly reaction was conducted exactly as described in Gibson et al. [53] and the products were electroporated into Top10F' *E. coli* cells (Invitrogen) and plated on selective plates. Clones were verified by PCR using the Prot fusion seq primer set in S1 Table, visualized through gel electrophoresis, and sequence verified. Plasmids were electroporated into *Asaia* SF2.1 electrocompetent cells, plated on selective media, and verified through PCR using the same primer set.

The MBP, GST, and TrxA fusion partner proteins were amplified from their respective plasmids using their matching primer pairs with homology to the Myc vector (Tables 1 and S1) and purified. The Myc vector backbone was digested with the *AvrII* restriction enzyme and recovered from gel electrophoresis after size confirmation with the Gel/PCR Fragment Extraction Kit (IBI Scientific, # IB47010). A two-fragment standard Gibson assembly procedure was again used for the individual insertion of the PCR fragments into the vector backbone [53]. The 6xHis double-stranded DNA fragment was achieved by annealing single-stranded complementary oligonucleotides together which were designed with *AvrII* and *AscI* overhanging ends for ligation into the Myc vector backbone (S1 Table). Briefly, the 500 μM

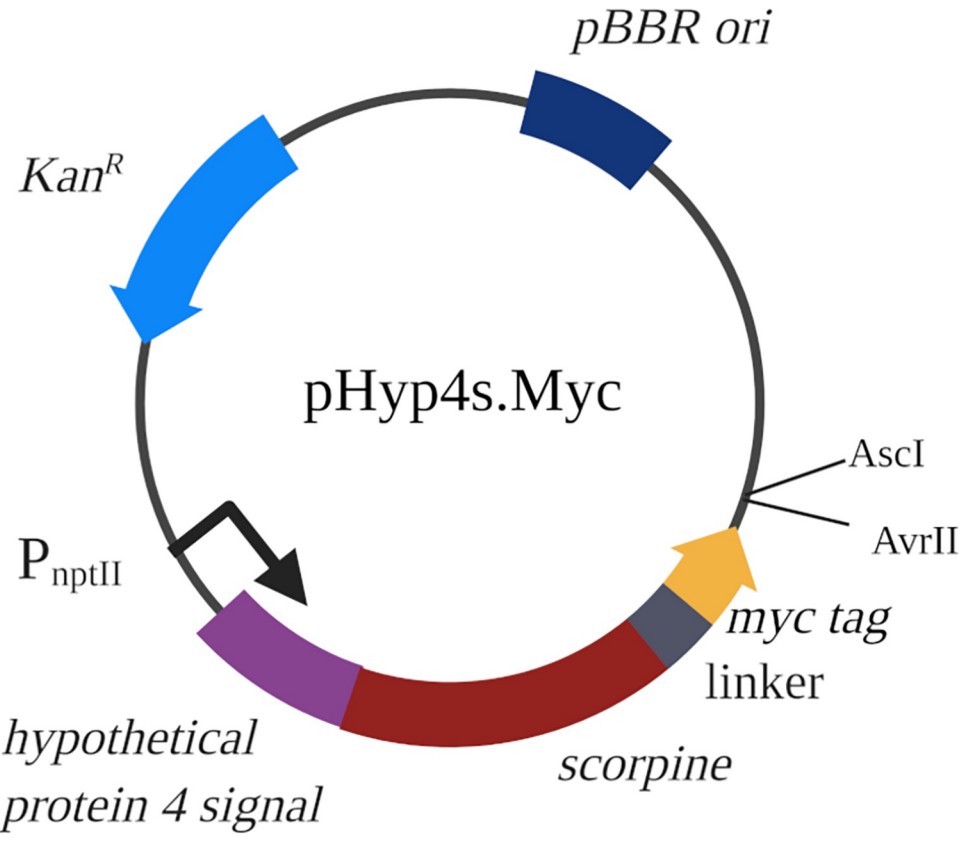

**Fig 1. Antiplasmodial vector pHyp4s.Myc.** This plasmid was created for fusion partner protein insertion by replacing the alkaline phosphatase region of Hyp4s (PhoA) [31] with a Myc tag and restriction enzyme recognition sites. The partner protein genes were inserted between the *AvrII* and *AscI* restriction sites for in frame fusion with the preceding Myc tag. pBBR, origin of replication; Kan^R, kanamycin resistance; $P_{nptII}$, promoter; hypothetical protein 4 signal, signal sequence for extracellular release of protein; scorpine, antiplasmodial effector protein; linker, $(GGGGS)_3$ flexible linker. Created with BioRender.com.

oligonucleotide stock solutions were diluted to a final concentration of 100 μM in deionized water. The oligonucleotides were mixed in equal molar amounts, along with 2 μl of 10× Duplex Buffer [0.98% potassium acetate, 0.72% HEPES (w/v), pH 7.5] and deionized water to bring the final volume to 20 μl. The mix was heated to 94°C for 2 min and allowed to gradually cool. The mix was diluted 1:1000 in deionized water and 3 μl was used for assembly into the digested Myc vector using a temperature-cycle ligation procedure [54]. The products were transformed via electroporation into *E. coli* Top10F' cells, PCR and sequence verified using the Prot fusion seq primer set in S1 Table. Plasmids were electroporated into *Asaia* SF2.1 electro-competent cells, plated on selective media, and verified through PCR using the same primer set.

## ELISA

*Asaia* strains expressing the fusion partner constructs were grown overnight. Cultures were streaked on mannitol plates and grown at 30°C for 48 h. Colonies were collected by flooding the plates with 1 mL of minimal media, gently scraping the cells from the plate, and collecting the liquid into centrifuge tubes. The $OD_{600}$ of each sample was taken, and all samples were diluted to reach a final $OD_{600}$ of 20. The diluted samples were then centrifuged at 13,800 x*g*

(12,000 RPM) for 5 min and the supernatants were collected in a separate tube. Two hundred microliters of the supernatant fraction were bound in wells of a NUNC-Immuno Maxisorp 96-well plate (VWR, cat. # 62409–024) overnight at 4°C. Plates were washed three times with Tris-buffered saline (TBS), and then blocked by adding 200 μl of 2% BSA-TBS (TBS with 2% (w/v) fraction V BSA) and incubating for 2 h at room temperature. The plate was again washed three times with TBS. Then, 100 μl of a 1:1000 dilution of mouse monoclonal anti-Myc antibody (Invitrogen, #46–0603) in 2% BSA-TBS was added to each well and incubated for 1 h at room temperature. The plate was washed eight times with 0.1% Tween20 (v/v)-TBS (TBS-T) for 2 min per wash. 100 μl of a 1:2000 dilution of goat anti-mouse-HRP antibody (Thermo-Fisher Scientific, #G-21040) in 2% BSA-TBS was added to each well and incubated for 1 h at room temperature, and the eight wash steps from above repeated. To visualize the protein, 50 μl of 1-Step Ultra TMB-ELISA (Thermo-Fisher Scientific, #34028) was added and the reaction allowed to proceed for 10–20 min at room temperature. To stop the reaction, 50 μl of 2 M $H_2SO_4$ was added, and the absorption at 450 nm was measured using a SpectraMax i3x plate reader (Molecular Devices), using the absorption at 655 nm for reference.

## Western analysis

Supernatant samples were collected and diluted in the same manner as above, except that MBP was diluted to a final $OD_{600}$ of 10 to prevent oversaturation on the blot. Seventy-five μl of each adjusted supernatant was added to 25 μl of 3x Laemmli buffer and the samples were boiled for 8 min. Fifteen μl of each sample and 8 μl of Precision Plus Protein™ Dual Xtra Pre-stained Protein Standards (BioRad cat. # 1610377) were loaded onto on an Any kD™ Mini-PROTEAN® TGX™ Precast Protein gel (BioRad cat. # 4569036) and separated at 200 V for 35 min. Proteins were then transferred onto a PVDF membrane in a BioRad transfer apparatus using Tris-Glycine transfer buffer (0.303% (w/v) Tris, 1.127% (w/v) glycine, 10% methanol (v/v)) at 100 V for 1 h. The membrane was dried overnight, briefly rinsed in methanol then water, then blocked with 50% v/v Odyssey blocking buffer in TBS (50% (v/v) fraction V LI-COR Odyssey Buffer with TBS) for 1 h at room temperature with agitation and rinsed three times in deionized water for 5 min. The membrane was incubated in the primary antibody solution containing a 1:3,000 mouse monoclonal anti-Myc antibody (Invitrogen, #46–0603) diluted in 50% v/v Odyssey blocking buffer in TBS-T overnight at 4°C with agitation. The next day, the membrane was washed three times for 10 min with TBS-T. It was incubated for 1 h at room temperature with agitation in the secondary antibody solution containing a 1:20,000 IRDye® 800CW goat monoclonal anti-mouse antibody (LI-COR, cat. # 925–32210) diluted in 50% v/v Odyssey blocking buffer in TBS-T with 0.01% w/v SDS. Two washes for 10 min with TBS-T was followed by one wash for 10 min with TBS. The membrane was visualized on an Odyssey FC dual mode imaging system (LI-COR) using the 800 nm infrared fluorescent detection channel for 2 min. Image Studio Software 5.0 (LI-COR) was used for blot visualization.

## Fitness assessments of *Asaia* strains

Four methods were used to assess the fitness of *Asaia* strains, mostly following procedures outlined in Shane et al. [34]. First, the cell viability of each strain was assessed using a Bacterial Viability Assay Kit (Abcam, #ab189818) following the manufacturer's procedure. *Asaia* strains were grown overnight, and cultures were streaked on mannitol plates and grown at 30°C for 48 h. Colonies were collected by flooding the plates with 1 mL of minimal media, gently scraping the cells from the plate, and collecting the liquid into centrifuge tubes. The $OD_{600}$ of each sample was taken, and all samples were diluted to reach a final $OD_{600}$ of 2 in 1 mL of minimal

media. The diluted samples were then centrifuged at 10,000 x*g* for 10 min, and the pellet was resuspended in 2 mL wash buffer. 1 mL of each suspension was added to 5 mL of wash buffer and incubated at room temperature for 1 h. Samples were centrifuged at 10,000 x*g* for 10 min, and the pellets were resuspended in 1 mL of wash buffer. One μl of the Total Cell Stain and 1 μl of the Dead cell stain were added to each sample, as well as a blank solution containing 1 mL wash buffer, and were incubated are room temperature in the dark for 1 h. Two hundred μl of each sample and the blank solution were placed in triplicate into NUNC flat bottom 96-well plate. Using a SpectraMax i3x (Molecular Devices) and Softmax Pro 7 software (Molecular Devices), fluorescence was measured at Ex 490nm/EM 525 nm (reading 1) and Ex 536nm/EM 617nm (reading 2) for all wells. The percentage of dead cells was calculated by dividing reading 2 by reading 1 and multiplying by 100 after blank correction.

Secondly, the maximum growth rate of each strain was measured. Each *Asaia* strain was inoculated at 0.1 $OD_{600}$ in 200 μl of a 96 well plate. The $OD_{600}$ was analyzed over 24 h at 15 min intervals using a SpectraMax i3x (Molecular Devices). SoftMax Pro 7 software (Molecular Devices) was used to create growth curves of collated replicates of each strain until they reached stationary phase. Growth curves were further analyzed using the package growth-rates59 [55] in RStudio to find the maximum growth rate of each strain of *Asaia*. Data was visualized in RStudio using boxplot.

The ability of the transgenic strains to survive when competed in a culture with wild-type *Asaia* was tested in a competition experiment. The antiplasmodial strains were competed with the wild type, ten replicates of each were grown to log phase (PhoA, Myc, TrxA, GST, 6xHis, MBP) and were individually mixed with wild-type *Asaia* SF2.1 in a 50/50 ratio of a 0.5 $OD_{600}$ culture. These mixed cultures were allowed to grow for 6 h, then diluted to $1.0 \times 10^{-6}$ $OD_{600}$ of which 100 μl was plated on mannitol media with or without 120 μg/ml kanamycin. The ratio of transgenic bacteria was calculated by comparing the colony forming units (CFUs) on the selective media to the CFUs on the nonselective media. Data was visualized in RStudio using boxplot.

The final fitness assessment that was performed was a mosquito colonization experiment that was performed in triplicate. Each *Asaia* strain was fed to female *An. stephensi* mosquitoes at a 0.1 $OD_{600}$ dilution in a sugar meal. After 36 h, mosquito midguts were dissected and homogenized using a tissue grinder. Fifteen midguts for each strain were pooled and diluted in 1000 μl of TBS. These samples were again diluted 10-fold in TBS and 100 μl of each dilution was plated on kanamycin and ampicillin supplemented minimal media with arabinose. CFUs for each strain were counted and compared to the total number of CFUs collected across test groups. Data was visualized in RStudio using boxplot.

### *Plasmodium berghei* parasite inhibition

The ability of the *Asaia* strains to inhibit *Plasmodium bergei* development in mosquitoes was evaluated according to Shane et al. [34]. Adult female ND4 Swiss Webster mice (21–24 g) were infected with *P. berghei* ANKA2.34 and parasites were allowed to develop in the mice until parasitemia level reached 4–10%, usually taking ca. 3 d. At this point the mice were sacrificed by placing them into a $CO_2$ chamber (30% displaced volume $min^{-1}$) and blood was collected via cardiac puncture. The infected blood was diluted with RPMI media (Gibco) to 2% parasitemia, then 200 μl ($5 \times 10^7$ parasites) was injected intraperitoneally into an uninfected mouse. At the time of this transfer, each *Asaia* scorpine strain to be tested was diluted to 0.1 $OD_{600}$ in the sugar meal and fed to 20–25 female *An. stephensi* mosquitoes in individual cups with screen lids. Thirty-six h post-infection each test group of mosquitoes was blood-fed on the infected mouse for 6 min each. Mice were anesthetized for this procedure by intraperitoneal injection

of 150 μl of the following mixture: {2 ml of 100 mg/ml Ketaject (Phoenix Pharmaceutical), + 1 ml of 10 mg/ml ACE Promazine (Henry Schein) + 7 ml saline} and placed on top of a mosquito cage at 19˚C under a warm cloth. The ability of the parasite to undergo exflagellation was also tested at this time using 6 μl ookinete media (1 L RPMI media supplemented with 0.2% sodium bicarbonate, 0.005% hypoxanthine, 0.00025% xanthurenic acid (w/v)) mixed with 10% (v/v) fetal bovine serum, 2 μl of 1 mg ml$^{-1}$ of heparin in sterile phosphate buffer (PBS) (0.8% NaCl, 0.02% KCl, 0.144% $Na_2HPO_4$, 0.024% $KH_2PO_4$, pH 7.2 (w/v)), and 2 μl of blood collected from a tail prick of the mouse. At least 2 exflagellation events occurred for each malarial trial. Exflagellation occurs when microgametes exit red blood cells after a female mosquito takes a *Plasmodium*-infected blood meal and can be monitored by microscopy. The number of these events in the blood meal is a measure of how infectious it is to the mosquito. After mosquito feeding, infected mice were sacrificed by placing them into a $CO_2$ chamber (30% displaced volume min$^{-1}$) until they stopped moving and voided their bladders, followed by thoracic bisection.

Mosquitoes that did not take in a blood meal were removed from the cups and discarded. Parasites were allowed to develop in the rest of the mosquitoes for 14 days at 19˚C in order to form oocysts. After 14 d, the mosquito midguts were dissected and stained with a 10-fold dilution of 1% (w/v) mercurochrome stain (Sigma Aldrich Product# M7011) in PBS for 30 min. They were then left to destain for 5 min in sterile PBS. The midguts were analyzed at 200X magnification and the number of oocysts per midgut was counted for each test group. All steps in this process were performed blindly and ordered randomly. Data was visualized in RStudio using Bee Swarm.

### Statistics and reproducibility

For all boxplots, the box bars are medians. The top and bottom of the boxes represent the first and third quartile of the data spread. The lower and upper bounds of the whiskers are the lowest datum still within 1.5X interquartile range (IQR) of the lower quartile, and the highest datum still within 1.5X IQR of the upper quartile, respectively. Significance for all tests was set to $P<0.05$. Variance was estimated using standard error of the mean and is appropriately similar between test groups of each experiment. Significance of the mean was calculated using one way ANOVA with Dunnett's correction in RStudio appropriate for multiple comparisons to a single control with normal distribution unless otherwise noted.

The data for the *Plasmodium berghei* inhibition analysis are pooled from four individual experiments. The median value of oocysts per midgut for the pooled data from the four experiments was calculated by and compared between treatments using quantile regression in RStudio [56]. Quantile regression is a non-parametric test that compares subsets of a data set individually and is useful for data showing unequal variation [56]. The significance of the difference in *P. berghei* oocyst prevalence was evaluated using binomial $\chi^2$ tests with 1 degree of freedom. All colony and oocyst counts were done blindly regarding which strain was evaluated, and the strains were ordered randomly. All data were deposited with datadryad.org at doi:10.5061/dryad.n2z34tmxw.

## Results

### Antiplasmodial effector release using various fusion partner proteins

A Gblock encoding a Myc epitope tag was added in frame after the linker to replace the alkaline phosphatase gene of the pHyp4s (PhoA) vector to create pHyp4s.Myc (Myc) (Fig 1). This vector was then used for construction of the other four fusion partner plasmids (Table 1). These fusion partner strains were tested for release of the recombinant protein outside of the cell.

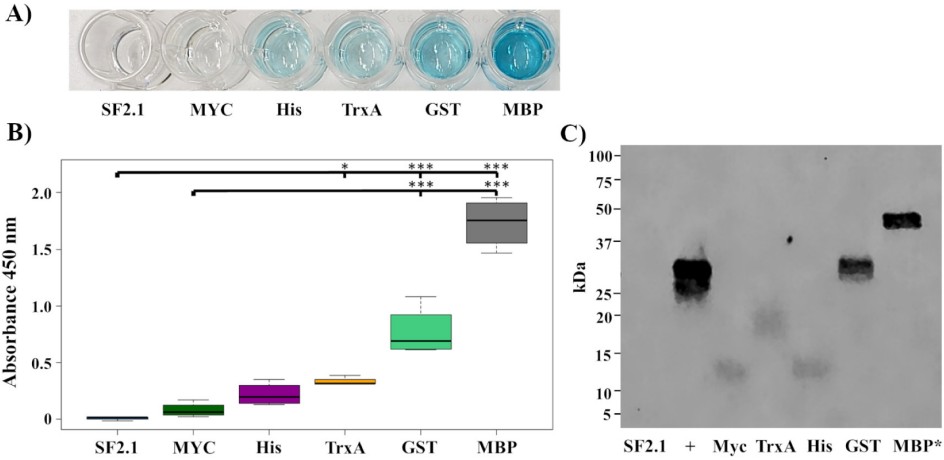

**Fig 2. Antiplasmodial effector abundance in the supernatants from *Asaia*.** (A) A representative ELISA that utilized anti-Myc primary and anti-mouse-HRP secondary antibodies to detect the presence of the recombinant protein in the supernatant fraction of transgenic *Asaia* strains. SF2.1, wild-type *Asaia*. (B) Quantification of supernatant fractions of the ELISA analysis. Relative levels (expressed as absorbance values measured at 450 nm) of substrate cleaved by the HRP-conjugate anti-mouse antibody across five separate trials. Statistical significance was determined using one-way ANOVA with Dunnett's correction where significance is represented by $^*P < 0.05$, $^{**}P < 0.01$, and $^{***}P < 0.001$ with experimental replicates. SF2.1, wild-type *Asaia*. (C) Representative western blot of supernatants from antiplasmodial *Asaia* strains. SF2.1, wild-type *Asaia*; +, pIT2-scFv Myc positive control; MBP*, MBP diluted to a final $OD_{600}$ of 10.

First, an ELISA was performed using the supernatants (Fig 2A). The supernatants from cells collected from plates were used and the protein of interest was detected using an anti-myc antibody and an anti-mouse-HRP conjugated antibody. While the protein of interest was detected for all five recombinant strains, only three strains (TrxA, GST, and MBP) were significantly different from the SF2.1 control (one way ANOVA with Dunnett's correction, $P \le 0.0152$), with GST and MBP showing the highest levels of protein detected in the supernatant (Fig 2B).

Western analysis was carried out on the supernatants of the new antiplasmodial strains as well as SF2.1 as a negative control and pIT2 as a Myc positive control using the anti-Myc antibody (Table 1). Except for the SF2.1 wild-type, a single band for our protein of interest was detected in all experimental lanes (Fig 2C), unlike our previous fusions to alkaline phosphatase which generates multiple bands when detected with an anti-PhoA antibody [31].

## Fitness assessments of recombinant antiplasmodial *Asaia* strains

Four fitness assessments were carried out on the *Asaia* strains to determine if any were at a disadvantage when compared to the *Asaia* SF2.1 strain. First, the cell viability of each strain on solid media was assessed (Fig 3A). When comparing the percentage of dead cells in each strain to the *Asaia* SF2.1 wild-type, only two of the strains, Myc and MBP, had significantly higher amounts (one way ANOVA with Dunnett's correction, $P \le 0.016$), with MBP having a mean value of 38.4% dead cells compared to a mean of 8.89% for SF2.1. The TrxA and GST strains had the lowest percentage of dead cells comparable with SF2.1 (one way ANOVA with Dunnett's correction, $P < 0.998$). MBP was the only strain to have a significantly higher percentage of dead cells when compared to the PhoA strain (one way ANOVA with Dunnett's correction, $P \le 0.001$).

Secondly, the maximum growth rates (μmax) of the strains in liquid culture were compared (Fig 3B). None of the transgenic strains grew as well as the *Asaia* SF2.1 wild-type strain (one

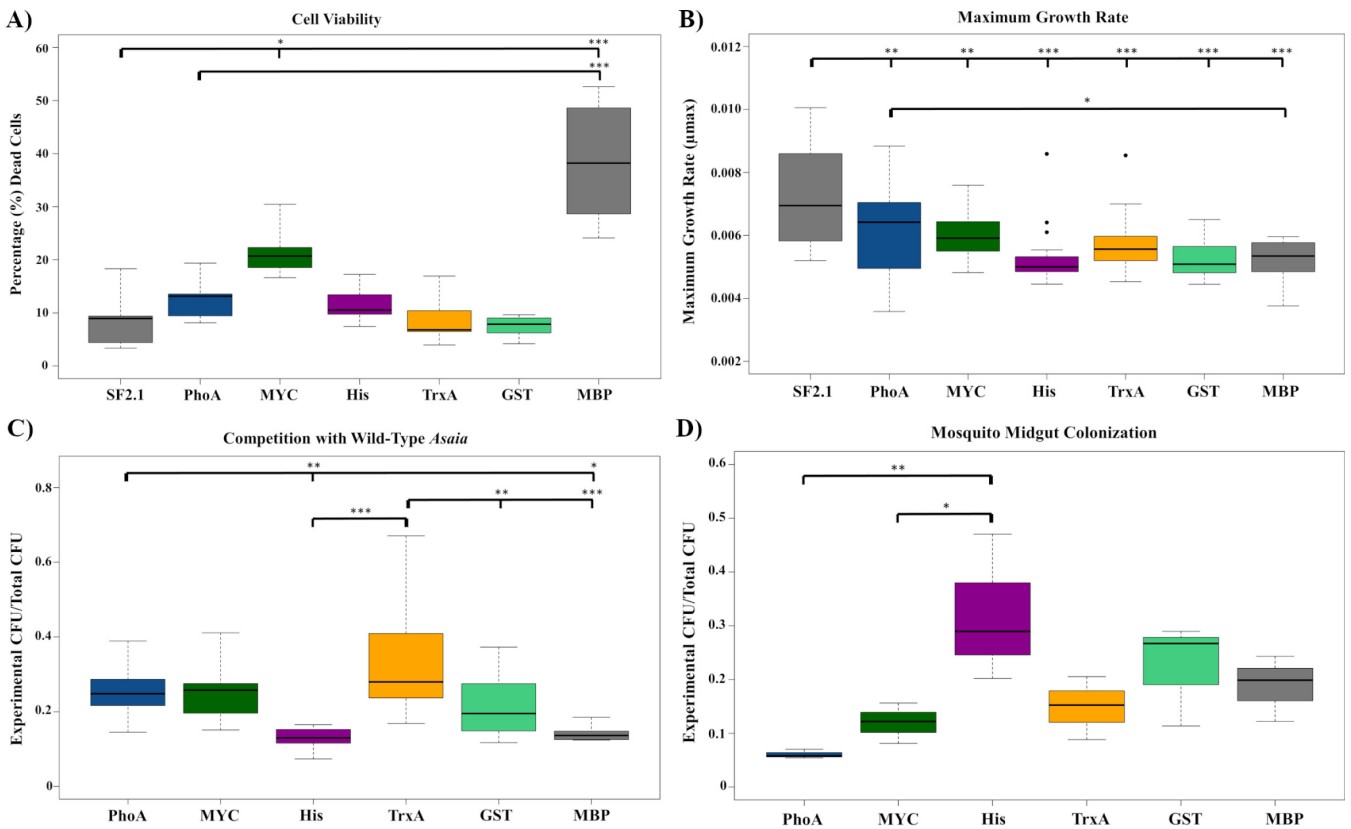

**Fig 3. Fitness assessments of antiplasmodial *Asaia* strains.** (**A**) Cell viability of *Asaia* strains on solid media. The Myc and MBP strains had the highest significant percentage of dead cells while SF2.1, TrxA and GST had the lowest percentage. (**B**) Growth curves and maximum growth rates (μmax) were calculated from individual isolates of each strain. Only MBP showed a significantly lower μmax than the PhoA strain. (**C**) Each strain was competed in mannitol media with wild-type *Asaia* SF2.1. Ratios of transgenic vs. whole culture CFUs are displayed. A 50:50 ratio indicates no loss of the paratransgenic strain during the course of the experiment. (**D**) Relative colonization of mosquito midguts by antiplasmodial *Asaia* strains. The His strain had the highest rate of colonization, followed by the GST strain; the PhoA strain had the lowest rate of colonization. Statistical significance for each experiment was determined using one-way ANOVA with Dunnett's correction where significance is represented by *$P < 0.05$, **$P < 0.01$, and ***$P < 0.001$ with experimental replicates (A. n ≤ 18, B. n ≤ 10, C. n = 3). Only significant comparisons are shown.

way ANOVA with Dunnett's correction, $P \leq 0.00316$). When the newly constructed strains were compared to a previously-developed PhoA strain, only MBP showed a significantly lower μmax (one way ANOVA with Dunnett's correction, $P = 0.03995$).

The third fitness assessment performed was a competition experiment in which each antiplasmodial strain was grown in a 50/50 co-culture with wild-type *Asaia* inoculated at a 0.5 OD$_{600}$, which is the beginning of log phase for the bacteria. These cultures were allowed to grow for 6 h and then plated on mannitol media with or without antibiotic. The ratio of surviving transgenic *Asaia* was calculated by comparing the number of colony forming units (CFUs) on the kanamycin supplemented plates, which only contained strains harboring the antiplasmodial plasmid, to the non-selective plates, containing all of the bacteria present in the culture (Fig 3C). The His and MBP strains showed the greatest reduction in the ratio of paratransgenic to wild-type CFUs when compared to PhoA (one way ANOVA with Dunnett's correction, $P \leq 0.01614$). The TrxA strain retained significantly higher ratios of transgenic bacteria compared to all other fusion strains except for Myc (one way ANOVA with Dunnett's correction, $P \leq 0.00541$).

The relative ability of the strains to colonize the mosquito midgut was the last assessment performed (Fig 3D). The transgenic strains were fed to mosquitoes, midguts were dissected and plated under conditions that selected for *Asaia* growth, and CFUs were counted across all strains. Interestingly, the His strain colonized the midgut significantly better than both the PhoA and Myc strains (one way ANOVA with Dunnett's correction, $P \leq 0.03230$), while there was no significant difference between PhoA and the other fusion strains ($P \geq 0.09233$).

## Activity of paratransgenic *Asaia* strains against *Plasmodium berghei*

The antiplasmodial strains of *Asaia* developed here were tested for their ability to prevent the development of *P. berghei* oocysts in *An. stephensi* female mosquito midguts, along with the Hyp4 (a strain releasing only alkaline phosphatase), PhoA, and SF2.1 strains as controls. The paratransgenic and wild-type strains were fed to female *An. stephensi* mosquitoes, which in turn fed on a *P. berghei* infected mouse. Mosquitoes that successfully blood-fed were dissected fourteen days after the infective blood meal, and oocysts per midgut were counted (Fig 4). All

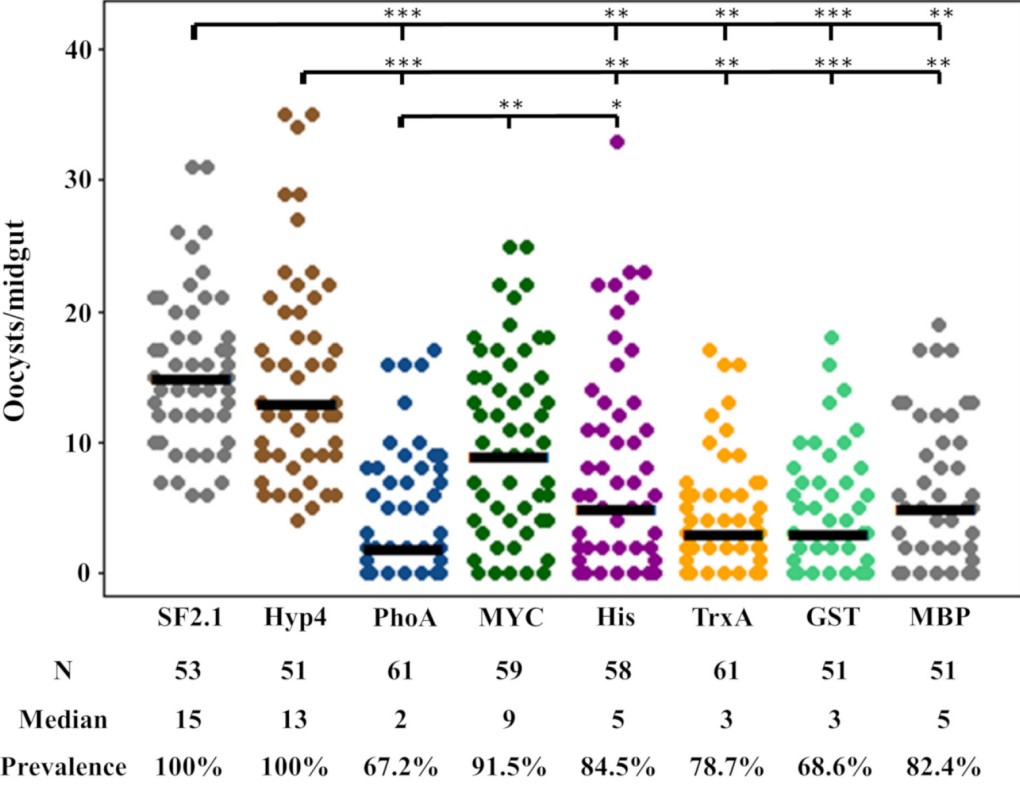

| | SF2.1 | Hyp4 | PhoA | MYC | His | TrxA | GST | MBP |
|---|---|---|---|---|---|---|---|---|
| N | 53 | 51 | 61 | 59 | 58 | 61 | 51 | 51 |
| Median | 15 | 13 | 2 | 9 | 5 | 3 | 3 | 5 |
| Prevalence | 100% | 100% | 67.2% | 91.5% | 84.5% | 78.7% | 68.6% | 82.4% |

**Fig 4. Suppression of *P. berghei* development by paratransgenic *Asaia* strains.** In four separate trials, oocysts were counted in mosquitoes infected with *Asaia* strains that were fed on a *P. berghei* infected mouse. Each dot represents an individual midgut and the number of *P. berghei* oocysts it contained. Prevalence is the fraction of midguts with at least one oocyst. SF2.1 is the wild type *Asaia* strain (negative control), Hyp4 is the *Asaia* strain releasing only alkaline phosphatase (negative control), and PhoA is a previously developed scorpine antiplasmodial strain with alkaline phosphatase as the fusion partner protein (positive control). All antiplasmodial strains significantly reduced the median number of oocysts (horizontal bars) compared to the wild-type strain (quantile regression, $P < 0.01$), while only the Myc and MBP antiplasmodial strains had a significantly higher median number of oocysts compared to the PhoA strain (quantile regression, $P < 0.02161$). The prevalence of infection was also significantly different between the wild-type and all antiplasmodial strains except for the Myc strain ($\chi^2$, 1 df). Only the Myc and His antiplasmodial strains had a significantly higher prevalence of infection compared to the PhoA strain ($\chi^2$, 1 df). There was no significant difference between the wild-type and Hyp4 negative controls. *P*-values: $^*P < 0.05$, $^{**}P < 0.01$, $^{***}P < 0.001$.

of the antiplasmodial strains significantly reduced the median number of oocysts when compared to the SF2.1 wild-type control *Asaia* strain (quantile regression, $P \leq 0.0024$), while the Hyp4 strain had a significantly higher median number of oocysts when compared to all but the wild-type and Myc strains ($P \leq 0.00965$). Myc and MBP had a significantly higher median oocyst count compared to the PhoA strain (quantile regression, $P \leq 0.02161$) while there was no difference between PhoA and the other three newly constructed strains ($P \geq 0.06734$).

Another measure of antiplasmodial activity is prevalence. Prevalence is the fraction of mosquitoes of a population that have at least one oocyst, meaning that they may still be infective and may be able to transmit the *Plasmodium* parasite. All of the antiplasmodial strains except for the Myc strain showed a significant reduction in prevalence of *P. berghei* infection (15.5%-32.8%) when compared to the wild-type and Hyp4 strains ($\chi^2$, $P \leq 0.008204$). The prevalence for three of the five new strains (TrxA, GST, and MBP) was not significantly different from the prevalence for the PhoA strain ($P \geq 0.1085$).

## Discussion

Many vector-borne diseases like malaria continue to plague humans across the globe, particularly in tropical areas [2]. Growing concerns over current preventative approaches and increased insect and parasite resistance highlight the need for new strategies [3–5]. One new proposed strategy focuses on genetically altering the mosquito genome itself for mosquito population suppression or to reduce vectorial capacity, but challenges such as reproductively isolated populations and large number of species that can vector the diseases may prove difficult [9–12]. One other proposed approach, paratransgenesis, can be used in conjunction with current strategies and may prove easier to implement due to ease of engineering the symbionts, ease of dispersal to natural populations, and lower fitness costs to the mosquito host itself [13–15, 35, 57, 58]. Our previous work investigated the ability of paratransgenic *Asaia* strains to inhibit *Plasmodium berghei* within the *Anopheles stephensi* mosquito midgut using different native signal sequences to release the antiplasmodial protein scorpine fused to alkaline phosphatase [31]. While infectivity rates were lowered using the antiplasmodial strains, there were some fitness costs to the bacteria host. Alkaline phosphatase is roughly 50 kDa. We used it previously as a reporter protein and it may not be necessary for antiplasmodial activity. Therefore, we hypothesized that by changing the fusion partner for the antiplasmodial effector, and especially reducing the length of the overall construct, we would be able to reduce the fitness costs to the *Asaia* host.

In this study, we evaluated five partner proteins fused to the C-terminal end of the antiplasmodial scorpine protein for their ability to mitigate toxicity to *Asaia*. These partners ranged in size from a small epitope tag (Myc, approx. 1.3 kDa) to a larger protein similar in size to alkaline phosphatase (MBP, approx. 40.2 kDa). Scorpine is a protein that shares properties of both cecropins and defensins and is a strong antimicrobial [33]. Even though its exact mechanism is not known, it has been shown to affect not only *Plasmodium* development in the mosquito midgut but also the fitness of *Asaia* when constitutively expressed [31, 32, 34]. Fusion proteins, which can help to enhance the folding of proteins, may increase solubility and protect the protein from intracellular degradation, and might also insulate the bacteria from the effects of scorpine.

When assessing the amount of protein released outside of the cell, GST and MBP released the most while the smaller Myc and 6xHis partners released the least (Fig 2). The Myc and 6xHis recombinant proteins are less than 20 kDa and intracellular degradation could also account for the low amounts of protein expressed from these strains [59]. It is believed that these recombinant proteins are not released by bona fide secretion but rather by disruption to

the OM, so it is also possible that these smaller constructs fail to disrupt the OM and thus fail to be released, as is seen with the other larger proteins [31]. Thioredoxin, GST, and MBP have been used extensively in industrial applications to increase the expression and solubility of proteins produced in bacterial hosts [42–44]. Hammarström et al. [60] compared the ability of seven different N-terminal proteins or tags to increase soluble expression of 27 small human proteins in *E. coli* and found that overall thioredoxin and MBP performed better than GST and 6xHis tag. Another study found GST to be better than MBP and 6xHis when fused to 32 human proteins of variable length [61]. MBP was ranked the best in two other studies when fused to different mammalian, plant, and insect proteins [62, 63]. Additionally, MBP was a better solubility enhancer when used as a C-terminal fusion tag in contrast to thioredoxin, and has been shown to have intrinsic chaperone activity that might actually reverse any misfolding of the preceding protein [43, 62]. One common conclusion from these studies, however, is that there is not a single fusion partner that works the best for every target protein.

The fitness assessments of the antiplasmodial strains provide insights into the fitness costs associated with increased protein expression. While MBP led to the greatest amount of protein released from the cell, it was the poorest performer in three out of the four fitness measurements when compared to both the wild-type and PhoA strains (Fig 3). With the highest percentage of dead cells and slower growth, the large amount of protein seen in Fig 2 is likely due to cell death and lysis rather than bona fide secretion. It is imperative that the paratransgenic strains not only express the protein in sufficient amounts, but that they also are able to survive in the mosquito midgut and compete with native bacteria. Under these criteria, the GST strain performed the best in terms of protein release from the cell and fitness of *Asaia*, followed by TrxA (Figs 2 and 3). In addition to their sensitivity to proteolytic degradation, antimicrobial proteins produced in microorganisms have innate toxic activity that could be harmful for the bacterial host. In industrial applications, fusion partner proteins have been utilized to not only increase the solubility of the recombinant proteins, but also to prevent the cytotoxicity of the protein on the host bacterium [59, 64]. We hypothesized this could be translated for use in paratransgenesis to counteract toxic effects from the antiplasmodials and to increase bacterial fitness, but the results were mixed in *Asaia* when using only one antiplasmodial, scorpine. It is possible that fusion of other antiplasmodials with these partner proteins could lead to greater bacterial fitness and parasite killing power, especially an effector specific for *Plasmodium*.

Finally, our mosquito experiments performed here showed that three strains of *Asaia* (TrxA, GST, and MBP) were able to decrease the prevalence of oocysts per midgut by 17.6 to 31.4%, which was not statistically different from the previously used PhoA strain that reduced the prevalence by 32.8% (Fig 4). This also shows that scorpine was still able to function properly against *Plasmodium* with the new fused partner proteins, possibly either due to non-interference from the fused partner or because it was released due to proteases activated during the bloodmeal ingestion [65–67]. Prevalence is the most important statistic to assess since even a single oocyst can cause a mosquito to become infectious. In nature, mosquitoes are usually infected with just 1–5 oocysts per midgut [9, 68]. Therefore, it is possible that these *Asaia* strains could perform even better in the field.

In conclusion, we were able to successfully evaluate five partner proteins fused to the antimalarial protein scorpine for their ability to mediate the fitness costs associated with heterologous protein production and extracellular release while still maintaining a high level of parasite killing power in the mosquito midgut. Since paratransgenic strains for field release would need to produce more than one effector protein to combat evolving parasite resistance, it is imperative to have multiple partners that can be combined with various antiplasmodials. This research brings *Asaia* one step closer for field-readiness and shows the attractiveness and

potential of paratransgenesis to combat not only malaria, but many other any vector-borne diseases.

## Supporting information

**S1 Table. Oligonucleotides and synthetic dsDNA fragments used in this study.** Restriction sites are underlined.
(PDF)

**S1 Raw image. Original uncropped western blot corresponding to Fig 2C in the manuscript, marked in the same manner.** The membrane was visualized on an Odyssey FC dual mode imaging system (LI-COR) using the 800 nm infrared fluorescent detection channel for 2 min. Image Studio Software 5.0 (LI-COR) was used for blot visualization. SF2.1, wild-type *Asaia*; +, pIT2-scFv Myc positive control; MBP*, MBP diluted to a final $OD_{600}$ of 10.
(PDF)

## Acknowledgments

We thank Guido Favia for the gift of *Asaia bogorensis* SF2.1 and Marcelo Jacobs-Lorena and the Johns Hopkins Malaria Research Center for the gift of *Anopheles stephensi*.

## Author Contributions

**Conceptualization:** Christina Grogan, David J. Lampe.

**Formal analysis:** Christina Grogan, David J. Lampe.

**Funding acquisition:** David J. Lampe.

**Investigation:** Christina Grogan, Marissa Bennett, David J. Lampe.

**Project administration:** David J. Lampe.

**Writing – original draft:** Christina Grogan.

**Writing – review & editing:** David J. Lampe.

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
