## [Decision Letter · Decision Letter 0]

10 Jul 2022

PONE-D-21-34954An evaluation of fusion partner proteins for paratransgenesis in Asaia bogorensisPLOS ONE

Dear Dr. Lampe,

Thank you for submitting your manuscript to PLOS ONE. After careful consideration, we feel that it has merit but does not fully meet PLOS ONE’s publication criteria as it currently stands. Therefore, we invite you to submit a revised version of the manuscript that addresses the points raised during the review process.

We look forward to receiving your revised manuscript.

Kind regards,

Yara M. Traub-Csekö

Academic Editor

PLOS ONE

Journal Requirements:

2. To comply with PLOS ONE submissions requirements, in your Methods section, please provide additional information on the animal research and ensure you have included details on (1) methods of sacrifice, (2) methods of anesthesia and/or analgesia, and (3) efforts to alleviate suffering 

3. We note that you have stated that you will provide repository information for your data at acceptance. Should your manuscript be accepted for publication, we will hold it until you provide the relevant accession numbers or DOIs necessary to access your data. If you wish to make changes to your Data Availability statement, please describe these changes in your cover letter and we will update your Data Availability statement to reflect the information you provide

Reviewers' comments:

Reviewer's Responses to Questions

**Comments to the Author**

1. Is the manuscript technically sound, and do the data support the conclusions?

Reviewer #1: Yes

Reviewer #2: Yes

Reviewer #3: Yes

2. Has the statistical analysis been performed appropriately and rigorously? 

Reviewer #1: Yes

Reviewer #2: Yes

Reviewer #3: Yes

3. Have the authors made all data underlying the findings in their manuscript fully available?

Reviewer #1: Yes

Reviewer #2: Yes

Reviewer #3: Yes

4. Is the manuscript presented in an intelligible fashion and written in standard English?

Reviewer #1: Yes

Reviewer #2: Yes

Reviewer #3: Yes

5. Review Comments to the Author

Reviewer #1: In this study, the authors evaluated five partner proteins fused to the C-terminal end of the antiplasmodial scorpine protein for their ability to mitigate toxicity to Asaia. The study is original, technically well conducted, and written in good English. The figures are of good quality and demonstrated the results obtained. Despite the article had been written in good English, some mistakes are present in the text. Below are some items that deserve revision in the text:

Line 16-

The form of the noun disease does not appear to be correct when used before the phrase such as. Consider changing the noun form to “diseases”.

Line 136- The word “herafter ” seems to be in the wrong spelling, missing the letter "e" before the letter "a". Change to “hereafter”.

Line 239- I believe that instead of “minimamedia ” the authors are referring to “minimal media”. Please fix it.

Line 234 and 237- “ … and incubated at room temperature for 1 hr . Samples were centrifuged at 10,000 xg for 10 min, …”

“… wash buffer, and were incubated are room temperature in the dark for 1 hr.”

The international symbol of hour is “h” not “hr”. The authors should change “hr” for “h” when referring to time lapses in hours.

Line 241- In the phrase: “The percentage of dead cells were calculated by dividing reading 2 by reading 1 and multiplying by 100 after blank correction.”

The verb were does not seem to agree with the subject. Consider changing the verb form to “was”.

Line 290- “The midguts were analyzed at 200X magnification and the number of oocysts per midgut were counted for each test group.” The verb were does not appear to agree with the subject the number of oocysts per midgut. Consider changing the verb.

Lines 369- 370 – “Secondly, the assessment of the maximum growth rates (µmax) of the strains in liquid culture were compared (Fig 3B).”

The plural verb were does not appear to agree with the singular subject assessment. Consider changing the verb form for subject-verb agreement.

Lines 386 – 387 – “The transgenic strains were fed to mosquitoes, midguts were dissected and plated under conditions that selected for Asaia growth, and CFUs were counted across all strains.”

It seems that you are missing a verb. Consider adding the verb “were”.

Line 434 – In the phrase “ …population suppression or to reduce vectoral capacity, but challenges such as reproductively..”

Correct the spelling : change “vectoral” to “vectorial”.

Reviewer #2: In this work, the bacterium Asaia, a symbiont found in insects, was transformed using a plasmid containing different combinations of DNA sequences including the scorpine fused with different partner proteins. This strategy has been tested to block the infection by Plasmodium, the etiological agent of malária. The expression of a heterologous protein, in this case, a toxin, is capable of inhibiting Plasmodium without causing damage to the mosquito and to the transgenic bacterium itself. Five different types of Asaia were transformed in different ways and these bacteria were compared considering different parameters under laboratory conditions. These different bacteria were compared regarding the level of protein expression, cell viability and ability to colonize the digestive system of Anopheles stephensi and tested for the ability to inhibit the development of Plasmodium berghei in the mosquito. The ability of the mosquito inoculated with the bacteria to transmit the parasite should have been tested to more accurately test the anti-plasmodial action of the bacteria, which was not the case. Furthermore, the passage of the bacteria in different generations of mosquitoes (vertical transmission) should have been studied to show which transgenic bacteria are more viable in interrupting the Plasmodium cycle.

Minors:

Indicate the insect families in the introduction (L65)

Indicate in the materials and methods the age of the mice used in the assays and how many days after infection (parasite passage) they were used to feed the mosquitoes.

Reviewer #3: Grogan and colleagues have conducted a set of experiments evaluating five different tags or partner proteins fused to the strong antimalarial effector molecule scorpine.

This paper is an interesting and useful contribution to the literature on the utility of using Asaia for the control of vector mosquitoes.

The manuscript is well written and scientifically sound and the experiments appear to be thoughtfully executed and the analysis is suitable. The authors are careful so as not to overstate their results.

Just minor comments or suggestions are listed below.

Abstract:

Line 22: correct "sp"; it must not be in italics.

Introduction:

line 53: correct "sp"; it must not be in italics.

Materials and Methods:

line 123: please indicate the strain of Anopheles stephensi.

line 181: replace ELISA with ELISA assays.

Line 189: specify the abbreviation

Line 194: one-hundred instead of 100ul

Line 229: minimal media

Line 234-237: 1h instead of 1hr. In the paragraph you have used 1h.

Line 241: was calculated

Line 254: a space is missing between 6 and h.

Line 256: specify the abbreviation

Discussion:

Line 511: the bibliography related to the use of Asaia for a paratransgenic application can be improved citing for example other studies (e.g. doi: 10.1038/s42003-020-0835-2.).

6. PLOS authors have the option to publish the peer review history of their article (what does this mean?). If published, this will include your full peer review and any attached files.

Reviewer #1: No

Reviewer #2: No

Reviewer #3: No

---

## [Author Response · Author response to Decision Letter 0]

28 Jul 2022

Reviewer 1: Reviewer 1 noted mostly grammatical problems with the manuscript which we appreciate having been caught. We accepted the majority of the suggested changes by this reviewer. Exceptions were the Line 369-70 comment where we removed the phrase “the assessment of” to bring the verb and subject into agreement and make the sentence less-awkward. Also, the phraseology of the Lines 386-87 sentence seems fine to us so we did not change it.

Reviewer 2: We added family designations to the species mentioned around Line 65. 

We added clarifications on mouse age and time after infection from around lines 273-276. We use mice based on weight and that is now noted in the manuscript. We infect mice twice for this procedure: once to produce “fresh” parasites and another time (in different mice) to produce infected mice for feeding by mosquitoes. That is now clarified (with timings) in the section on “Plasmodium berghei parasite inhibition”.

Reviewer 2 suggested that the correct way to determine the effect of our Asaia strains with regard to the transmission of Plasmodium berghei would be to test their ability to block transmission from mosquitoes to mice. We acknowledge that this is the most precise way to measure the antiplasmodial effect of our strains, however those experiments are costly to perform, especially with regard to the number of mice necessary. In principle, each infected mosquito would be allowed to bite a single mouse, and then that mouse would be measured for parasite transmission. This kind of experiment would involve dozens of mice. That experiment might be justified for strains of Asaia that are in an advanced stage of development, but not for a survey of passenger proteins and their effects on parasite killing in mosquitoes. The measurement of oocyst numbers and prevalence is the standard measure in this field for experiments like the ones we performed here. Indeed, the measure of prevalence is a conservative measure. Mice that have no oocysts cannot transmit P. berghei. Mice that have one oocyst might conceivably be impaired for transmission by the paratransgenic strains, but we count those as being infectious. Therefore, prevalence is an upper limit on the effect of our strains and we talk about it that way. Full-blown transmission to mice is rarely done, for exactly the reasons we outlined above.

Reviewer 2 also suggested that we perform experiments that measure the transmission of our strains from one mosquito generation to the next to measure persistence. We certainly agree that persistence of our strains in the field will be an important component to measure to predict the overall success of a paratransgenesis. Again, those experiments are premature at this stage where we are simply trying to measure performance in mosquitoes with regard to parasite killing. The lab strains we have built here are not suitable for release into the field since they carry drug markers and their plasmids may be subject to horizontal transfer. These are issues we are addressing in other studies.

Reviewer 3: Reviewer 3 noted mostly grammatical problems which we appreciate being caught. We accepted most of these changes with the following exceptions:

• The strain of An. stephensi we use is the one under cultivation at the Malaria Research Center at Johns Hopkins U. We received these as a gift from Marcelo Jacobs-Lorena directly from the center’s insectary. Dr. Jacobs-Lorena had no other strain data or reference to pass along to us when we contacted him about them other than that they originated at NIH >30 years ago. Given this situation, we did not change the wording of the sentence at line 123 since we already indicated that they came from Johns Hopkins.

• We left “ELISA” the way it is written. ELISA is an acronym for Enzyme Linked Immunosorbant Assay. To add the word “assay” to ELISA seems redundant to us.

• We left “100 ul” as it was and chose not to spell out “one hundred”. It is our understanding of general usage that numbers over ten are normally not spelled out.

• We added the reference that was suggested.

---

## [Decision Letter · Decision Letter 1]

11 Aug 2022

An evaluation of fusion partner proteins for paratransgenesis in Asaia bogorensis

PONE-D-21-34954R1

Dear Dr. Lampe,

We’re pleased to inform you that your manuscript has been judged scientifically suitable for publication and will be formally accepted for publication once it meets all outstanding technical requirements.

Kind regards,

Yara M. Traub-Csekö

Academic Editor

PLOS ONE

Additional Editor Comments (optional):

Reviewers' comments:

Reviewer's Responses to Questions

**Comments to the Author**

1. If the authors have adequately addressed your comments raised in a previous round of review and you feel that this manuscript is now acceptable for publication, you may indicate that here to bypass the “Comments to the Author” section, enter your conflict of interest statement in the “Confidential to Editor” section, and submit your "Accept" recommendation.

Reviewer #1: All comments have been addressed

Reviewer #2: All comments have been addressed

Reviewer #3: All comments have been addressed

2. Is the manuscript technically sound, and do the data support the conclusions?

Reviewer #1: Yes

Reviewer #2: Yes

Reviewer #3: Yes

3. Has the statistical analysis been performed appropriately and rigorously? 

Reviewer #1: Yes

Reviewer #2: Yes

Reviewer #3: Yes

4. Have the authors made all data underlying the findings in their manuscript fully available?

Reviewer #1: Yes

Reviewer #2: Yes

Reviewer #3: Yes

5. Is the manuscript presented in an intelligible fashion and written in standard English?

Reviewer #1: Yes

Reviewer #2: Yes

Reviewer #3: Yes

6. Review Comments to the Author

Reviewer #1: (No Response)

Reviewer #2: (No Response)

Reviewer #3: (No Response)

7. PLOS authors have the option to publish the peer review history of their article (what does this mean?). If published, this will include your full peer review and any attached files.

Reviewer #1: No

Reviewer #2: No

Reviewer #3: No

---

## [Editor Report · Acceptance letter]

23 Aug 2022

PONE-D-21-34954R1 

An evaluation of fusion partner proteins for paratransgenesis in *Asaia bogorensis*

Dear Dr. Lampe:

I'm pleased to inform you that your manuscript has been deemed suitable for publication in PLOS ONE. Congratulations! Your manuscript is now with our production department. 

Kind regards, 

on behalf of

Dr. Yara M. Traub-Csekö 

Academic Editor

PLOS ONE